# Sociodemographic factors associated with the success or failure of anti-tuberculosis treatment in the Chiapas Highlands, Mexico, 2019–2022

Héctor Javier Sánchez-Pérez[1,2,3,4]☯*, Cristina Gordillo-Marroquín[1,2,3,4]☯*, Janeth Vázquez-Marcelín[5], Miguel Martín-Mateo[4,6,7], Anaximandro Gómez-Velasco[2,4,8]

1 Departamento de Salud, El Colegio de la Frontera Sur, San Cristóbal de Las Casas, Chiapas, México, 2 Red Mexicana de Investigación en Tuberculosis y otras Micobacteriosis A.C. (REMexTB), Ciudad de México, México, 3 Observatorio Social de Tuberculosis de México, México, 4 Grupos de Investigación para América y África Latinas (GRAAL), Barcelona, España, 5 Programa de Prevención y Control de la Tuberculosis del Distrito de Salud-San Cristóbal, Secretaría de Salud del Estado de Chiapas, San Cristóbal de Las Casas, Chiapas, México, 6 Facultad de Medicina, Universidad Autónoma de Barcelona, Cerdanyola del Vallès, España, 7 Universidad Internacional del Ecuador, Quito, Ecuador, 8 Departamento de Ecología Humana, Instituto Politécnico Nacional, Cinvestav, Mérida, Yucatán, México

☯ These authors contributed equally to this work.
* cmgordillo@ecosur.mx (CGM); hsanchez@ecosur.mx (HJSP)

## Abstract

### Objective

To estimate the incidence rate of tuberculosis (TB) in the Highlands (Tsotsil-Tseltal) region of Chiapas and to analyze sociodemographic factors that might influence the success of anti-TB treatment from the period of January 2019 to June 2022.

### Methods

Retrospective study in which the TB databases of the National Epidemiological Surveillance System (SINAVE) were analyzed. TB incidence rates were calculated based on the number of registered TB cases and estimated annual populations. The success-failure of anti-TB treatment was analyzed according to sociodemographic indicators, degree of concentration of indigenous population of the municipality of residence and admission to SINAVE.

### Results

Two hundred thirty-three cases were analyzed. The variables associated to a lower success rate of treatment against TB were: living in a municipality with high-very high concentration of indigenous population, being indigenous, having a primary school education or lower, and agricultural occupation. The number of TB diagnosed from 2020–2022 and the incidence rates from 2020–2021 decreased significantly compared to 2019.

**Data Availability Statement:** All relevant data are within the manuscript and its Supporting Information files.

**Funding:** The study was carried out with support from Partners in Health Sucursal Perú and CONAHCYT through the Cristina Gordillo-Marroquín postdoctoral fellowship (I1200/320/2022). The funders did not participate in any way in the design of the study, the collection and analysis of data, the decision to publish the work, or the preparation of the manuscript.

**Competing interests:** The authors have declared that no competing interests exist.

## Conclusions

It is necessary to strengthen the follow-up of TB cases in the region, mainly in areas with high-very high indigenous concentration, in people with low levels of education and engaged in agricultural work.

## Introduction

Tuberculosis (TB) is a communicable disease ranked as the second cause of death by a single infectious agent which has only been surpassed by the COVID-19 pandemic. In 2021 it was estimated that 10.1 million people contracted TB worldwide (with an interval between 9.9 and 11.0 million), and that 1.6 million died because of this, a number recorded in 2017, meaning that in 2021 there was a setback of at least four years [1].

In 2022, according to estimates of the National Center for Disease Prevention and Control (CENAPRECE), the incidence rate of TB in Mexico was of 18.5 per 100,000 inhabitants. In that year, 26,463 cases were reported, from which 23,449 and 588 were new cases and drug TB resistant, respectively [2]. Chiapas, a southern Mexican state, held the 11˚ place with a rate of 27.0 cases in the same denominator. However, this figure may be underestimated due to the high levels of underdiagnosis in the region [3]. Regarding to national TB mortality rate registered in the country for 2021, it was 1.3 per 100,000 inhabitants–with 2,292 notified deaths-, and the state of Chiapas ranked 11[th] with a rate of 1.8 cases in the same denominator [2, 4].

As in most of the world, the scant progress achieved in TB control up to 2019 in Mexico was hindered due to the COVID-19 pandemic. There were severe impacts- recognized by the National Tuberculosis Program (NTP)- on the prevention activities, diagnosis, and case follow-up (routinely carried out Direct Observed Treatment Short-Course, DOTS), which led to a rise in TB deaths in the country and worldwide in 2020 [1, 5].

Various studies have shown the high TB burden in Indigenous People [6–10]. In Mexico, the status of TB in this vulnerable population is unknown, as well as the impact of the COVID-19 pandemic on the prevention, diagnosis, treatment, follow-up, and cure of TB cases.

TB thrives in vulnerable populations such as Indigenous People. In Mexico, around 6.1% speak an indigenous language (7,364,645 people) [11]. In Chiapas, 32.7% of its population is indigenous, which usually lives in rural areas or in the outskirts of cities. Their living conditions contribute to burden of TB: overcrowded and poor quality housing, food insecurity, barriers to access health services, and low level educational background, among other issues [12]. In this sense, Chiapas is one of the states with higher levels of poverty in the country. In 2018, it was estimated that 78% of its population was experiencing poverty, out of which 47% was in moderate poverty and 30.6% in extreme poverty [13].

The heightened socioeconomic vulnerability of the indigenous populations in Chiapas, in addition to factors such as migration, loss of indigenous language and culture [14], and lately, insecurity and social violence [3], significantly contribute to their exclusion and to perpetuate and increase inequities and inequalities in the health of Indigenous People.

Furthermore, the emergence of COVID-19 resulted in drawbacks for TB prevention and control programs. There was reassignment of healthcare workers (doctors, nurses, lab technicians, etc.), as well as physical, material, and economic resources to face the pandemic. Health workers that were considered as a risk group due to age or having a comorbidity, were released from their duties contributing to shortage of staff. On the other hand, in the general population there was a fear of contracting COVID-19, with consequences for not seeking medical attention despite having symptoms related to TB. All the above factors made access to health

services more difficult for the most vulnerable groups, which were more pronounced in Indigenous People [15, 16].

The emergence of COVID-19 also disrupted vaccination campaigns, trigger distrust and misinformation concerning vaccines [3], leading to a reduced immunization coverage, where the Peninsula Region (which includes Campeche, Chiapas, Quintana Roo, Tabasco, and Yucatan) had the lowest percentage (19.8%) of complete immunization coverage (including BCG) in children of up to 2 years of age in 2021, while national coverage was of 31.1% [17].

It is to be expected that all these factors may increase the risk of contracting and developing TB when there is not an opportune diagnosis (which perpetuates the spread cycle and infection in communities) to provide early treatment. Likewise, the diagnosis of TB in rural areas, where access to health services is scarce or virtually nonexistent, is complicated due to the clinical spectrum of the disease [18], given that, generally, in marginalized areas only the most advanced TB cases are identified, because the main diagnostic method available is still smear microscopy [19].

The objective of this study was to analyze the situation of TB in the Altos (Tsotsil-Tseltal) region of Chiapas, which has a high proportion of native population, during the period January 2019 to June 2022, as well as to analyze sociodemographic factors that might influence the success of the anti-tuberculosis treatment.

## Methods

A retrospective cross-sectional observational study was performed. All TB cases registered in the TB Platform of the National Epidemiological Surveillance System (SINAVE) from January 1, 2019 to June 30, 2022 in the Tsotsil-Tseltal region of Chiapas, Mexico, were analyzed. The data were accessed for research purposes on May 16, 2023 (after the research protocol was approved by the corresponding research ethics committee). To protect the anonymity and confidentiality of individuals, once duplicate records were identified and removed, all personal information (name, national identification number, city of residence, and address) was deleted prior to analysis. Only one of the authors (JVM) had access to all the data.

TB incidence rates of 2019, 2020, 2021 and 2022 were calculated with the number of registered TB cases as the numerator and the estimated population at a municipal level for each year of study as denominator [20].

The response variable, "success or failure of anti-tuberculosis treatment", was categorized according to the classification used by the National Tuberculosis Program (NTP) [21]: Being cured and having finished the anti-tuberculosis treatment were considered as "success", while "failure" included loss to follow-up, abandonment, failure, and death from TB. To analyze the success or failure of anti-TB treatment, the following variables were studied:

a) Demographics: age, sex, ethnic affiliation (indigenous, non-indigenous, according to whether the registry established that the person spoke or not a native language of the region), concentration of indigenous population in the municipality of residence: low-medium (less than or equal to 50%) and high-very high (more than 50%) [22, 23].

b) Socioeconomics: schooling (up to primary, secondary or higher level), occupation (agricultural, non-agricultural).

c) Clinics: type of TB, pulmonary or extrapulmonary; program admission classification, new or not new case (relapse, readmission due to abandonment or failure); time elapsed between the onset of symptoms, seeking care, and diagnosis of TB; drug resistance; comorbidities (diabetes, nutritional status, HIV, alcoholism, smoking, and hypertension).

d) Year of diagnosis: 2019, 2020, 2021, and 2022. The chosen period is because 2019 was the pre-COVID-19 pandemic period and because in that year the current TB data collection

platform began to be used by the NTP. On the other hand, June 2022 was established as the end date of the research because, when the study was carried out in 2023, the registered cases of TB should have had completed their anti-tuberculosis treatment to be included in the corresponding analysis.

Results are presented using the test of comparison of proportions (Chi-square), likelihood ratio, and the Student's T test (in the case of quantitative variables) at a significance level of $p < 0.05$. Finally, a step-by-step logistic regression model was also performed to analyze the possible association of the variables studied that showed an association with the response variable. The SPSS (IBM) software package version 21 was used for the statistical analysis.

### Ethical approval

The research protocol was approved on May 15, 2023, by the Research Ethics Committee of El Colegio de la Frontera Sur (The College of Southern Border, CEI-030523). Individual written consent was not applicable because this was a retrospective study of clinical data (record review).

## Results

### Characteristics of the study population

In the period analyzed, 233 people affected by TB (PATB) were registered, 87 (37.3%) from 2019, 40 (17.2%) from 2020, 58 (24.9%) from 2021 (during these 3 years, from January to December) and, 48 (20.6%) in 2022 (from January to June). Demographic and sociodemographic characteristics of the studied population, as well as some indicators of entry and exit to the SINAVE are shown in Table 1.

### TB incidence rates

Taking as a reference the number of TB cases in 2019, the reduction of cases in 2020 and 2021 was of 51.7% and 33.3%, respectively. Hence, the incidence rate of TB in 2020 decreased 55.2%, with a slight increase in 2021, without reaching the pre-pandemic level of COVID-19 pandemic (Table 2).

### Success of anti-tuberculosis treatment

As shown in Table 1, out of the 233 people registered with TB during the period studied, 175 (75.1%) were considered by the Tuberculosis Prevention and Control Program as success cases (cured or at least finished their anti-tuberculosis treatment). Forty people (17.1%) were considered failure-lost to follow-up, 6 (2.6%) died from TB, 10 (4.3%) died from other causes, and 2 (0.9%) were still under treatment (both registered in 2022).

### Bivariate analysis

There was not statistical difference between the success or failure of treatment according to the age of the persons studied. However, women had a higher percentage of treatment success than men (85.6% vs. 74.8%), a difference statistically significant under maximum likelihood ratio ($p < 0.05$). Being indigenous and living in a municipality with a high-very high concentration of indigenous population was associated with a lower percentage of anti-tuberculosis treatment success (Table 3).

Regarding education, people with secondary or more level had a higher percentage of treatment success than people that only studied up to primary school. Concerning occupation,

**Table 1. Demographic and socioeconomic characteristics of the studied population (n = 233) registered with tuberculosis in the Highlands Region V Tsotsil-Tseltal of Chiapas, Mexico, during the period January 2019 to June 2022.**

| Sociodemographic indicators | | |
|---|---|---|
| Age (in number of years) | Mean (standard deviation) | 39.6 (16.9) |
| | Median | 36 |
| Sex (%) | Woman | 39.9 |
| | Man | 60.1 |
| Ethnicity (%) | Indigenous (Tsotsil and Tseltal) | 79.4 |
| | Non-indigenous | 20.6 |
| Concentration of indigenous population in the municipality of residence (%) | High–very high | 67 |
| | Middle–low | 33 |
| Schooling (% at the level reached) | Up to primary | 72.8 |
| | Secondary or higher | 27.2 |
| Occupation (% type of occupation) | Agricultural | 36.1 |
| | Housework, none | 41.7 |
| | Other | 22.2 |
| **SINAVE entry and exit indicators** | | |
| Type of Tuberculosis (%) | Pulmonary TB | 94.4 |
| | Nodal | 1.7 |
| | Other | 3.9 |
| New case (%) | Yes | 94.4 |
| | No | 5.6 |
| Final classification of anti-TB treatment (%) | Cured–end of anti-TB treatment | 75.1 |
| | Failure, loss to follow-up | 17.1 |
| | Death from TB | 2.6 |
| | Death from other causes | 4.3 |
| | In anti-TB treatment | 0.9 |

TB: Tuberculosis. Source: Own elaboration based on TB cases registered in the SINAVE and in the indicators of the degree of marginalization by municipality, 2020 of the National Population Council based on the INEGI Population and Housing Census 2020 [24].

**Table 2. Tuberculosis incidence rates per 100,000 inhabitants in the Highlands Region V Tsotsil-Tseltal of Chiapas, Mexico, period 2019–2021.**

| Year | Number of registered TB cases | Estimated total population in the region | Tuberculosis incidence rate per 100,000 inhabitants |
|---|---|---|---|
| **2019** | 87 | 716,098 | 12.14 |
| **2020** | 40 | 772,321 | 5.17 |
| **2021** | 58 | 747,812 | 7.76 |
| **2022 (only January-June)** | 48 | (...) | (...) |
| **Average (2019–2021)** | 61.7 | 745,410 | 8.36 |

TB: Tuberculosis; (...): missing information.

Source: Own elaboration based on TB cases registered in the SINAVE and in the INEGI Population and Housing Census 2020

**Table 3. Percentages of success and non-success of anti-tuberculosis treatment in people affected by tuberculosis in the Highlands Region V Tsotsil-Tseltal of Chiapas, Mexico, period 2019–2022[a].**

| Indicator | % of success (cured-end of treatment) | % of non-success (failure, TB death, loss to follow-up) |
|---|---|---|
| **Sex[b]** | | |
| •Woman (n = 90) | 85.6 | 14.4 |
| •Man (n = 131) | 74.8 | 25.2 |
| **Ethnicity[b]** | | |
| •Indigenous (n = 178) | 76.4 | 23.6 |
| •Non-indigenous (n = 43) | 90.7 | 9.3 |
| **% of concentration of indigenous population[b]** | | |
| •High—very high (n = 151) | 74.8 | 25.2 |
| •Middle—low (n = 70) | 88.6 | 11.4 |
| **Schooling[b]** | | |
| •Up to primary (n = 161) | 75.8 | 24.2 |
| •Secondary or higher (n = 60) | 88.3 | 11.7 |
| **Occupation[b]** | | |
| •Agricultural (n = 79) | 70.9 | 29.1 |
| •Non agricultural (n = 140)[c] | 83.6 | 16.4 |
| **Year of TB diagnosis** | | |
| •2019 (n = 83) | 80.7 | 19.3 |
| •2020 (n = 35) | 85.7 | 14.3 |
| •2021 (n = 58) | 70.7 | 29.3 |
| •2022 (n = 45) | 82.2 | 17.8 |

Source: Own elaboration based on the results presented.

[a] For all analyses, two cases that were still under anti-tuberculosis treatment and ten cases of deaths due to causes other than tuberculosis were excluded.

[b] p<0.05

[c] Two cases for which no information was obtained were excluded.

people who do not engage in agricultural work had a higher percentage of success in their anti-tuberculosis treatment than those who do engage in this type of activity (Table 3).

With reference to clinical indicators, no differences were found in the success or failure of anti-tuberculosis treatment in relation to type of TB (pulmonary and extrapulmonary). According to admission category to the SINAVE, although there was a higher percentage of success in new cases (80.3%, n = 208) than in non-new ones (61.5%, n = 13), the difference was not statistically significant. Similarly, no differences were found in terms to the time elapsed between the onset of symptoms and seeking care and then getting a TB diagnosis: the average time was very similar between those who successfully completed their anti-tuberculosis treatment and those who did not (around three months for both groups). A similar situation occurred when the same variables were compared, grouping the data between people affected with tuberculosis living in municipalities with a high-very high and medium-low indigenous concentration.

Regarding the drug-resistant condition, only 4 mono-resistant cases were registered (3 to isoniazid and 1 to rifampicin, all 4 cases with treatment success).

For registered comorbidities, no differences were found in the percentage of treatment success or failure in the cases of diabetes (n = 35), malnutrition (n = 16), HIV (n = 6), alcoholism (n = 11), smoking (n = 8) and hypertension (n = 4).

According to the year of diagnosis of TB cases, the differences in success percentages were not statistically significant. The year with less success of treatment was 2021 and the year with greater success was 2020, although it was also the year with the lowest number of registered TB cases (Table 3).

## Multivariate analysis

We made two analyses: the first one, separating men and women, we found, by means of likelihood analysis, a significant association (with a value of 4.46, $p < 0.035$), showing that indigenous women have a 5.1 times higher anti-tuberculosis treatment failure rate than non-indigenous (19.0 vs. 3.7%, respectively). This effect was not observed in the case of men, in which there was no significant association.

When fitting a logistic regression model with all factors studied, using the step-by-step technique, only the indigenous population concentration of the municipality was statistically significant: OR = 2.61 (CI95% 1.14–5.95), which means that the ratio of proportions indicates 2.18 times more failures of the anti-tuberculosis treatment in zones with high-very high indigenous concentration.

## Discussion

The studied region is one of the 15 economic regions of Chiapas. It has a population of 755,821 inhabitants distributed in 17 municipalities [25], out of which 15 have a high-very high indigenous concentration. According to United Nations Development Program (UNDP) data, Chiapas has the lowest Human Development Index (HDI) in the country and, in turn, some of the municipalities in the studied region not only have the lowest HDI in the state but also in the country [26].

The population registered with TB during the time of the study shows clearly vulnerable demographic and socioeconomic characteristics: 8 out of every 10 people are indigenous, mostly living in municipalities with a high concentration of Indigenous People, slightly more than 7 out of 10 barely reached the primary school level, and almost 4 out of 10 work in agriculture, which is consistent with the fact that TB is closely related to the social determinants of health [27].

In Chiapas, Indigenous People usually live under conditions of overcrowding, poverty, high levels of malnutrition, scarce access to health services, and little knowledge of the disease, which makes them more susceptible to contracting and developing TB [28, 29], and having difficulties to comply with the anti-tuberculosis treatment, even when DOTS is provided in the region, just as the results of this study show.

Additionally, the control of TB is a constant challenge, given the economic, geographical, political, cultural, language and insecurity barriers that they face to obtain adequate TB care (prevention, diagnosis, and treatment), even more so when we consider human rights, gender, and intercultural care [19].

In this regard, three aspects of the incidence rates of TB estimated for each of the analyzed years (2019–2021) can be highlighted. Firstly, the drop observed in 2020 compared to 2019 stands outs, and then the slight increase in 2021, that did not reach the level of 2019 (Table 2). This is most likely due to the severe negative impact of the COVID-19 pandemic on the region, where: (i) great part of healthcare workers were moved to aid during the pandemic (or were part of the COVID-19 at risk groups, or got sick), so in spite of the strategies implemented to follow-up on TB cases through DOTS (even via telecommunication when possible), TB prevention, diagnosis and control actions were affected; (ii) people went less to health units, given government instructions to reduce social mobility (to avoid the spread of COVID-19) and for

fear of getting infected; (iii) healthcare workers were not allowed to enter some communities, not only out of fear, but also because of the rejection of the anti-COVID vaccine when it began to be administered, among other aspects [3, 16, 30].

Secondly, the spatial-temporal dynamic distribution of TB may also be determined by structural factors of the region, such as: (i) insufficient coverage, and, low quality and scarce physical and geographical access to health services, mainly in areas of greater marginalization and social exclusion, with ensuing low rates of TB diagnosis; (ii) cultural aspects, including language and different perceptions of the health-disease process [31–33]; (iii) the socioeconomic situation of the communities, combined with the existing political and insecurity conflicts, which are elements that promote the displacement and migration in search of better job opportunities, contributing to people settling in unsanitary conditions that eventually lead to development of TB, as well as less access to health services, which in turn makes it more difficult for them to be properly diagnosed and treated [34].

Thirdly, another element contributing to the low TB incidence rates found is that the people registered with TB represent the segment of the population that is in less unfavorable socioeconomic conditions and that does have access to TB diagnosis and treatment in the health services -that could explain that the failure-loss to follow-up rate is relatively low (17.1%) compared to other studies in deprived populations of Latin America [35, 36]. However, various community based studies carried out in the region have documented high levels of TB underdiagnosis [37, 38], as well as high mortality because of this disease [31].

Regarding to the percentage (75.1%) of cases with successful treatment–cured and with completion of treatment- (Table 1), it is low if we consider that the NTP itself defines a minimum of 85% [21]. Out of the analyzed factors that could identify the reason behind this, it can be mentioned that: (i) according to biological sex, women had a higher percentage of success than men (85.6% vs. 74.8%). It is possible that certain work and mobility issues in males, such as the incompatibility schedules with first-level health services and moving to other locations–whether temporary or permanent- could make it difficult for them to follow-up and continue their treatment; (ii) although there is a higher success rate of anti-tuberculosis treatment in women than in men, in women there are notable differences according to whether or not they are indigenous: the failure rate of indigenous women is 5.1 times higher than that of non-indigenous women (19.0 vs. 3.7%, respectively), effect not observed in men; and, (iii) being indigenous and living in a municipality of high-very high Indigenous People concentration is associated to less success in treatment in relation to their peers (Table 3), a finding that coincides with previous studies that have shown less non-compliance with the anti-tuberculosis treatment due to conditions of poverty and exclusion, such as low levels of education and low paying jobs, like agricultural work (elements that were also consistent in this study) [34], as well as health services scarcity, distance to them, lack of money to go to the corresponding health unit [39], negative perception that causes distrust of health services, and also, inter and intracommunity conflicts [19, 38].

Regarding clinical indicators, it can be pointed out that the type of TB (pulmonary–nonpulmonary) was not statistically associated with the success/non-success of treatment; this finding may be due to the small number of extrapulmonary cases (less than 10%). Similarly, both the analysis of new/not-new cases (208 vs. 13) and the resistance to anti-tuberculosis treatment (217 sensitive cases vs. four mono-resistant ones), were not statistically significant.

On the other hand, contrary to the expected, the time elapsed between the onset of symptoms and seeking care and then getting a TB diagnosis was not associated with greater or lesser treatment success, even after analyzing between people with TB as indigenous or non-indigenous. However, other studies have shown that there may be more than a 60% delay in the diagnosis and treatment of TB in Indigenous People compared to non-indigenous people [9]. In this context, the results of this study may stem from the fact that, at least in this population, the

search for care, diagnosis and success of anti-tuberculosis treatment depend more on individual, community and health services [40] structural factors, such as: knowledge about TB, type of primary healthcare agent (traditional healer), self-treatment or not seeking care, distance to the nearest health unit, transportation limitations, migration (internal and external), and language limitations, among others [9, 34, 40].

Another aspect to consider is the clinical spectrum of TB. A person may be infected by the bacillus and not develop the characteristic signs and symptoms of TB (latent TB and subclinical state). During these stages, the person will not seek medical care. However, these stages may precede the development of active TB [18, 41], especially in people in a deficient socioeconomic context, where usually the only available diagnosis method is the bacilloscopy, in spite of the existence, but not availability, of other more sensitive methods (culture and molecular tests) [1].

In relation to the influence of the presence of comorbidities on treatment outcome, there were also a limitation in terms of the small number of people registered with TB who had one or more comorbidities. As mentioned in the results section, only 35 of diabetes, 6 of HIV, 16 of malnutrition, 4 of hypertension, 11 of alcoholism and 8 of smoking were registered, which prevented analyzing their effect on the treatment outcome in the studied population.

Finally, the differences in the percentage of treatment success according to the year of registration of the person with TB did not reach statistical significance. Although the highest percentage of successful treatment (85.7%) was found in 2020, it was also the year in which fewer cases were diagnosed and treated (n = 35), while 2021 was the year with less success (70.7%, n = 58), which indicates that the effects of COVID-19 could be observed in the number of cases diagnosed in the region, but not necessarily on the success -or otherwise- of the treatment, that could be associated with the so-called social determinants of health.

In this sense, generating TB prevention and control interventions with extensive knowledge of the sociocultural, economic, and epidemiological situation of indigenous populations will be decisive in the control and eventual reduction of TB cases [10].

## Study limitations

Our study had the following limitations. Firstly, the study design was retrospective based on the collection of secondary data (SINAVE clinical records), so its accuracy could have been affected during the COVID-19. Secondly, considering the high underdiagnosis of TB in the country as a whole, particularly in the study region [37, 38], our results show only the situation of people with TB who have access to health services and, therefore, they do not illustrate the situation of people who do not attend health units in the region. They only represent a subgroup of the population affected by this public health problem and not all of them.

Thirdly, in this study it was not possible to analyze 10 cases of deaths registered for causes other than TB in SINAVE. They were excluded because health personnel did not know the causes and time of occurrence. It would have been very useful to conduct an analysis to identify the social, health and access to health services conditions in which these deaths occurred.

Carrying out population-based studies (i.e. household surveys) would give a closer approximation of the current TB situation in the studied region. Moreover, given that the used information was anonymous and confidential, it was not possible to corroborate/complete demographic, socioeconomic, and clinical information of people with TB, which would have been useful to perform a deeper analysis.

## Conclusions

The COVID-19 pandemic negatively impacted TB diagnosis in which the incidence rates drop (55.2%) from 2020–2021 compared to 2019. It is necessary to strengthen the follow-up of TB

cases in the region, mainly in areas with high-very high indigenous concentration, indigenous women, people with low levels of education and those engaged in agricultural work.

## Supporting information

**S1 Data.**
(XLSX)

## Author Contributions

**Conceptualization:** Héctor Javier Sánchez-Pérez, Cristina Gordillo-Marroquín.

**Data curation:** Héctor Javier Sánchez-Pérez, Cristina Gordillo-Marroquín.

**Formal analysis:** Miguel Martín-Mateo.

**Funding acquisition:** Héctor Javier Sánchez-Pérez, Cristina Gordillo-Marroquín.

**Investigation:** Héctor Javier Sánchez-Pérez, Cristina Gordillo-Marroquín, Janeth Vázquez-Marcelín, Miguel Martín-Mateo, Anaximandro Gómez-Velasco.

**Methodology:** Héctor Javier Sánchez-Pérez, Cristina Gordillo-Marroquín.

**Project administration:** Héctor Javier Sánchez-Pérez, Cristina Gordillo-Marroquín.

**Resources:** Janeth Vázquez-Marcelín.

**Supervision:** Héctor Javier Sánchez-Pérez, Cristina Gordillo-Marroquín.

**Validation:** Héctor Javier Sánchez-Pérez, Cristina Gordillo-Marroquín, Anaximandro Gómez-Velasco.

**Visualization:** Héctor Javier Sánchez-Pérez, Cristina Gordillo-Marroquín.

**Writing – original draft:** Héctor Javier Sánchez-Pérez, Cristina Gordillo-Marroquín.

**Writing – review & editing:** Héctor Javier Sánchez-Pérez, Cristina Gordillo-Marroquín, Janeth Vázquez-Marcelín, Miguel Martín-Mateo, Anaximandro Gómez-Velasco.

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
