## [Decision Letter · Decision Letter 0]

12 Nov 2023

PONE-D-23-25816The situation of tuberculosis of indigenous populations in Chiapas, Mexico, during 2019-2022 Tuberculosis in indigenous populationsPLOS ONE

Dear Dr. Gordillo-Marroquín,

Thank you for submitting your manuscript to PLOS ONE. After careful consideration, we feel that it has merit but does not fully meet PLOS ONE’s publication criteria as it currently stands. Therefore, we invite you to submit a revised version of the manuscript that addresses the points raised during the review process.

 Please submit your revised manuscript by Dec 27 2023 11:59PM. If you will need more time than this to complete your revisions, please reply to this message or contact the journal office at plosone@plos.org. Please include the following items when submitting your revised manuscript:A rebuttal letter that responds to each point raised by the academic editor and reviewer(s). You should upload this letter as a separate file labeled 'Response to Reviewers'.A marked-up copy of your manuscript that highlights changes made to the original version. You should upload this as a separate file labeled 'Revised Manuscript with Track Changes'.An unmarked version of your revised paper without tracked changes. You should upload this as a separate file labeled 'Manuscript'.If applicable, we recommend that you deposit your laboratory protocols in protocols.io to enhance the reproducibility of your results. Protocols.io assigns your protocol its own identifier (DOI) so that it can be cited independently in the future. For instructions see: https://journals.plos.org/plosone/s/submission-guidelines#loc-laboratory-protocols. Additionally, PLOS ONE offers an option for publishing peer-reviewed Lab Protocol articles, which describe protocols hosted on protocols.io. Read more information on sharing protocols at https://plos.org/protocols?utm_medium=editorial-email&utm_source=authorletters&utm_campaign=protocols.

We look forward to receiving your revised manuscript.

Kind regards,

Laura Soledad Lamfre, M.D

Academic Editor

PLOS ONE

Journal Requirements:

2. Please amend either the title on the online submission form (via Edit Submission) or the title in the manuscript so that they are identical.

Additional Editor Comments:

The analysis of the incidence of TB disease and its social determinants of health is very important for collective health. However there are some corrections that should be done for publication.

- The manuscript could incorporate, to add robustness, a more complex statistical analysis, as suggested by reviewer 2.

- You should correct the title, as suggested by reviewer 1.

- Add a coment about the strategics of the health system to guarantee TB treatment, as suggested by reviewer 2.

Reviewers' comments:

Reviewer's Responses to Questions

**Comments to the Author**

1. Is the manuscript technically sound, and do the data support the conclusions?

Reviewer #1: Partly

Reviewer #2: Yes

2. Has the statistical analysis been performed appropriately and rigorously? 

Reviewer #1: No

Reviewer #2: Yes

3. Have the authors made all data underlying the findings in their manuscript fully available?

Reviewer #1: Yes

Reviewer #2: Yes

4. Is the manuscript presented in an intelligible fashion and written in standard English?

Reviewer #1: No

Reviewer #2: Yes

5. Review Comments to the Author

Reviewer #1: The authors bring an interesting topic about Tb in a local region from Mexico.

However many there are many concerns that made the manuscript not suitable to be published in this journal.

- First of all, the informations are related only with a local region from Mexico, so a local jornal would be more appropriated to share the data.

- In the title the authors mention about TB in indigenous populations, however the data shown that although indigenous were the majority of the cases, others ethnicity were also present.

- The introduction section is too long making reading the text quite tiring.

- The results are repeated im many sections from the manuscript.

- The phrases were long and many times not properly structured.

Reviewer #2: The article is focused on a very important Collective Health problem: TB and social determinants, with a lens on one of most socioeconomically deprived regions of Mexico

The manuscript is technically sound, and the data support the conclusions. One comment: It is mentioned, but it should probably be highlighted: If there is evidence on a high rate of underdiagnosis of TB in the region, then the participants of this study may represent a subgroup, a selection of all the patients with TB and not all of them. Moreover, they may probably have better social determinants of health that are related to better access to treatment. That could explain that the failure-loss to follow-up rate is relatively low (17,1%) compared to other studies in deprived populations of LatinAmerica.

Another comment:There is almost no mention on the strategies of the health system to guarrantee TB treatmet: Is DOT routinelly done?

The statistical analysis been performed appropriately and rigorously. Perhaps other complementary statistical analysis could potentially offer other insigths. For example a logistic regression (where the dependent variable could be success/failure of treatment, and the independent variables could be all the sociodemographic and clinical information from the SINAVE database). Another interesting statistical approach could be a multi-level or Hierarchical analysis, in order to study the interactions of individual level data (being indigenous, language, educational level, occupation, etc), household level data (if available: water supply, electricity, gas, quality of house construction materials, etc) and municipality level data (average poverty, indigenous population, and other indicators).

The authors made all data underlying the findings in their manuscript fully available

The manuscript presented in an intelligible fashion and written in standard English. One single correction on table of sociodemographic indicators: Replace"mediana" by "median".

Congratullations to the authors for such an important study.

6. PLOS authors have the option to publish the peer review history of their article (what does this mean?). If published, this will include your full peer review and any attached files.

Reviewer #1: No

Reviewer #2: **Yes: **Santiago Hasdeu

---

## [Author Response · Author response to Decision Letter 0]

12 Dec 2023

Response to Reviewers_PlosONE

Journal Requirements:

Answer: Done.

2. Please amend either the title on the online submission form (via Edit Submission) or the title in the manuscript so that they are identical.

Answer: Done.

Answer: Done

Additional Editor Comments:

The analysis of the incidence of TB disease and its social determinants of health is very important for collective health. However there are some corrections that should be done for publication.

4. The manuscript could incorporate, to add robustness, a more complex statistical analysis, as suggested by reviewer 2.

Answer: We appreciate reviewer´s suggestion and we have carried out complementary statistical analyses to make our results more robust.

A) In the Methods section, we describe additional statistical analyses to provide more strength to our study (likelihood ratio, bivariate and multivariate analyses). 

Lines 167-172-. “Results are presented using the test of comparison of proportions (Chi-square), likelihood ratio, and the Student’s T test (in the case of quantitative variables) at a significance level of p<0.05. Finally, a step-by-step logistic regression model was also performed to analyze the possible association of the variables studied that showed an association with the response variable. The SPSS (IBM) software package version 21 was used for the statistical analysis”.

B) In the Results section, we describe the outputs for each statistical analyses carried out: 

Lines 209-214. 

“Bivariate analysis.

There was not statistical difference between the success or failure of treatment according to the age of the persons studied. However, women had a higher percentage of treatment success than men (85.6% vs. 74.8%), a difference statistically significant under maximum likelihood ratio (p<0.05)….”

Lines 251-261. 

“Multivariate analysis 

We made two analyses: the first one, separating men and women, we found, by means of likelihood analysis, a significant association (with a value of 4.46, p<0.035), showing that indigenous women have a 5.1 times higher anti-tuberculosis treatment failure rate than non-indigenous (19.0 vs. 3.7%, respectively). This effect was not observed in the case of men, in which there was no significant association.

When fitting a logistic regression model with all factors studied, using the step-by-step technique, only the indigenous population concentration of the municipality was statistically significant: OR = 2.61 (CI95% 1.14-5.95), which means that the ratio of proportions indicates 2.18 times more failures of the anti-tuberculosis treatment in zones with high-very high indigenous concentration”.

5. You should correct the title, as suggested by reviewer 1.

Answer: We appreciate reviewer´s observation. We have changed the original title and the new version has the following title: Sociodemographic factors associated with the success or failure of anti-tuberculosis treatment in Chiapas Highlands, Mexico, 2019-2022 

6. Add a comment about the strategics of the health system to guarantee TB treatment, as suggested by reviewer 2.

Answer: In the new version, in the Introduction and discussion sections, we stated the following:

Introduction

Lines 76-81. “….As in most of the world, the scant progress achieved in TB control up to 2019 in Mexico was hindered due to the COVID-19 pandemic. There were severe impacts- recognized by the National Tuberculosis Program (NTP)- on the prevention activities, diagnosis, and case follow-up (routinely carried out Direct Observed Treatment Short-Course, DOTS), which led to a rise in TB deaths in the country and worldwide in 2020 [1,5].”

Discussion

Lines 278-279. “….and having difficulties to comply with the anti-tuberculosis treatment, even when DOTS is provided in the region, just as the results of this study show”. 

Lines 290-292. “…so in spite of the strategies implemented to follow-up on TB cases through DOTS (even via telecommunication when possible), TB prevention, diagnosis and control actions were affected;”

Authors' responses to reviewers' comments and questions

We thank the reviewers for their thoughtful comments and time.

Reviewer #1: 

The authors bring an interesting topic about Tb in a local region from Mexico.

However many there are many concerns that made the manuscript not suitable to be published in this journal.

Answer: We are grateful for your comments and hope that the new version of the article will have addressed your observations and suggestions for modifications to the text. More importantly, in this study we analyzed and described the epidemiological situation of TB in Indigenous People who account for a large proportion of TB cases worldwide. Understanding the mechanisms to this situation, TB programs worldwide may benefit from such studies to propose strategies to improve TB prevention and control in those vulnerable populations and Ethnic minorities.

- First of all, the informations are related only with a local region from Mexico, so a local jornal would be more appropriated to share the data.

Answer: Despite being a study located in a region of Mexico with a significant presence of indigenous populations, TB in this vulnerable population has not been sufficiently analyzed both at the local and global levels. In Mexico, at least 12% of the population is indigenous (taking the native language as an indicator, so that the figure, if other indicators are considered, such as clothing, customs and traditions, self-ascription, would probably be much higher).

In more than 90 countries, some 476 million people identify themselves as belonging to indigenous peoples -representing 5% of the world's population-(https://www.amnesty.org/en/what-we-do/indigenous-peoples/), of which at least 42 million live in Latin America (https://www.worldbank.org/en/region/lac/brief/indigenous-latin-america-in-the-twenty-first-century-brief-report-page).

In most countries of Latin America and the Caribbean region, being indigenous is no longer a demographic indicator, and has been associated with poverty, social exclusion, stigmatization, and discrimination - including in health services-.

For this reason, studying the TB problem is essential. The contribution of this study lies in the fact that it makes visible the probabilities of successfully or unsuccessfully completing anti-tuberculosis treatment, depending on whether one lives in a municipality with a high or low indigenous concentration, as well as, in the case of women, whether one is indigenous or not.

Hence, we have chosen PLOS ONE due to its broad and more diverse audience to disseminate our results. Therefore, we are sure that, if published in PLOS ONE, the findings obtained will be useful not only for Mexico, but also for the great majority of Latin American countries with indigenous populations, which will most likely serve as an input to make visible and promote collective interest in the TB situation in these populations with greater vulnerability and, consequently, for the development of public policies in this regard.

- In the title the authors mention about TB in indigenous populations, however the data shown that although indigenous were the majority of the cases, others ethnicity were also present.

Answer: We appreciate reviewer´s suggestion and we changed the title of the study. The new version has the following title: Sociodemographic factors associated with the success or failure of anti-tuberculosis treatment in Chiapas Highlands, Mexico, 2019-2022 

- The introduction section is too long making reading the text quite tiring.

Answer: Thank you for your comment. The introduction section was revised and shortened, as suggested by the reviewer. 

- The results are repeated in many sections from the manuscript.

Answer: In the new version, we have made the necessary changes in the wording of the text to avoid the repetitions mentioned above.

- The phrases were long and many times not properly structured.

Answer: We appreciate reviewer´s observation. The new version of this article was reviewed by an English-speaking proofreader. 

Reviewer #2: 

The article is focused on a very important Collective Health problem: TB and social determinants, with a lens on one of most socioeconomically deprived regions of Mexico. The manuscript is technically sound, and the data support the conclusions. 

One comment: It is mentioned, but it should probably be highlighted: If there is evidence on a high rate of underdiagnosis of TB in the region, then the participants of this study may represent a subgroup, a selection of all the patients with TB and not all of them. 

Answer: We agree with reviewer´s observation. In the previous version, we stated the following:

“The results of the study only show the situation of people with TB who have access to health services and, therefore, they do not illustrate the situation of people who do not attend health units in the region.”

We have rewritten the above paragraph and in the new version we include the following: 

Lines 385-390. “….Secondly, considering the high underdiagnosis of TB in the country as a whole, particularly in the study region [37,38], our results show only the situation of people with TB who have access to health services and, therefore, they do not illustrate the situation of people who do not attend health units in the region. They only represent a subgroup of the population affected by this public health problem and not all of them.”

Moreover, they may probably have better social determinants of health that are related to better access to treatment. That could explain that the failure-loss to follow-up rate is relatively low (17,1%) compared to other studies in deprived populations of Latin America. 

Answer: Thank you for your comment. In the new version we have modified lines 310-317, as follows:

“Thirdly, another element contributing to the low TB incidence rates found is that the people registered with TB represent the segment of the population that is in less unfavorable socioeconomic conditions and that does have access to TB diagnosis and treatment in the health services -that could explain that the failure-loss to follow-up rate is relatively low (17.1%) compared to other studies in deprived populations of Latin America [35,36]-. However, various community based studies carried out in the region have documented high levels of TB underdiagnosis [37,38], as well as high mortality because of this disease [31]”

Another comment: There is almost no mention on the strategies of the health system to guarantee TB treatment: Is DOT routinelly done?

Answer: In the new version, in the sections of Introduction and discussion, we stated:

In the Introduction section:

Lines 76-81. As in most of the world, the scant progress achieved in TB control up to 2019 in Mexico was hindered due to the COVID-19 pandemic. There were severe impacts- recognized by the National Tuberculosis Program (NTP)- on the prevention activities, diagnosis, and case follow-up (routinely carried out Direct Observed Treatment Short-Course, DOTS), which led to a rise in TB deaths in the country and worldwide in 2020 [1,5]

In the Discussion section:

Lines 278-279. “….and having difficulties to comply with the anti-tuberculosis treatment, even when DOTS is provided in the region, just as the results of this study show”. 

Lines 290-292. “…so in spite of the strategies implemented to follow-up on TB cases through DOTS (even via telecommunication when possible), TB prevention, diagnosis and control actions were affected;”

The statistical analysis been performed appropriately and rigorously. Perhaps other complementary statistical analysis could potentially offer other insights. For example a logistic regression (where the dependent variable could be success/failure of treatment, and the independent variables could be all the sociodemographic and clinical information from the SINAVE database). 

Answer: Thank you for reviewer´s suggestion. We have included the following complementary statistical analyses to make our results more robust

A) In the Methods section, we describe additional statistical analyses to provide more strength to our study (likelihood ratio, bivariate and multivariate analyses). 

Lines 167-172-. “Results are presented using the test of comparison of proportions (Chi-square), likelihood ratio, and the Student’s T test (in the case of quantitative variables) at a significance level of p<0.05. Finally, a step-by-step logistic regression model was also performed to analyze the possible association of the variables studied that showed an association with the response variable. The SPSS (IBM) software package version 21 was used for the statistical analysis”.

B) In the Results section, we describe the outputs for each statistical analyses carried out: 

Lines 209-214. 

“Bivariate analysis.

There was not statistical difference between the success or failure of treatment according to the age of the persons studied. However, women had a higher percentage of treatment success than men (85.6% vs. 74.8%), a difference statistically significant under maximum likelihood ratio (p<0.05)….”

Lines 251-261. 

“Multivariate analysis 

We made two analyses: the first one, separating men and women, we found, by means of likelihood analysis, a significant association (with a value of 4.46, p<0.035), showing that indigenous women have a 5.1 times higher anti-tuberculosis treatment failure rate than non-indigenous (19.0 vs. 3.7%, respectively). This effect was not observed in the case of men, in which there was no significant association.

When fitting a logistic regression model with all factors studied, using the step-by-step technique, only the indigenous population concentration of the municipality was statistically significant: OR = 2.61 (CI95% 1.14-5.95), which means that the ratio of proportions indicates 2.18 times more failures of the anti-tuberculosis treatment in zones with high-very high indigenous concentration”.

Another interesting statistical approach could be a multi-level or Hierarchical analysis, in order to study the interactions of individual level data (being indigenous, language, educational level, occupation, etc), household level data (if available: water supply, electricity, gas, quality of house construction materials, etc) and municipality level data (average poverty, indigenous population, and other indicators).

Answer: Thank you for the reviewer's suggestions. At the individual level, the SINAVE database only includes age, sex, ethnicity, schooling, and occupation. As the reviewer will appreciate, these variables were included in the study.

Unfortunately, the SINAVE does not record other information requested by the reviewer (water supply, electricity, gas, quality of house construction materials, etc), which, had it been available, it would have enriched the analyses performed.

Finally, regarding data at the municipality level (percentage of poverty, indigenous population, and other indicators), we have included the only one available for study purposes: the concentration of indigenous population. In this regard, it should be noted that this indicator clearly reflects the poverty and marginalization conditions of the municipalities in the region studied. 

The authors made all data underlying the findings in their manuscript fully available.

The manuscript presented in an intelligible fashion and written in standard English. 

One single correction on table of sociodemographic indicators: Replace "mediana" by "median".

Answer: Done (Table 1)

Congratulations to the authors for such an important study.

Answer: Thank you very much for your very good remarks and comments.

---

## [Editor Report · Decision Letter 1]

26 Dec 2023

Sociodemographic factors associated with the success or failure of anti-tuberculosis treatment in the Chiapas Highlands, Mexico, 2019-2022

PONE-D-23-25816R1

Dear Dr. Gordillo-Marroquín,

We’re pleased to inform you that your manuscript has been judged scientifically suitable for publication and will be formally accepted for publication once it meets all outstanding technical requirements.

Kind regards,

Laura Soledad Lamfre, M.D

Academic Editor

PLOS ONE
---

## [Editor Report · Acceptance letter]

17 Jan 2024

PONE-D-23-25816R1 

PLOS ONE

Dear Dr. Gordillo-Marroquín, 

I'm pleased to inform you that your manuscript has been deemed suitable for publication in PLOS ONE. Congratulations! Your manuscript is now being handed over to our production team.

Kind regards, 

on behalf of

Lic. Laura Soledad Lamfre 

Academic Editor

PLOS ONE